# Second Victims Among Austrian Nurses (SeViD-A2 Study)

**DOI:** 10.3390/healthcare12202061

**Published:** 2024-10-17

**Authors:** Eva Potura, Hannah Roesner, Milena Trifunovic-Koenig, Panagiota Tsikala, Victoria Klemm, Reinhard Strametz

**Affiliations:** 1The Second Victim Association Austria, 1190 Vienna, Austria; 2Wiesbaden Institute for Healthcare Economics and Patient Safety (WiHelP), Wiesbaden Business School, RheinMain University of Applied Sciences, 65183 Wiesbaden, Germany; 3Training Center for Emergency Medicine (NOTIS e.V), 78234 Engen, Germany

**Keywords:** second victim, emotional distress, nurses, psychological distress, patient safety, occupational health

## Abstract

**Background:** The Second Victim Phenomenon (SVP) significantly impacts the well-being of healthcare professionals and patient safety. While the SVP has been explored in various healthcare settings, there are limited data on its prevalence and associated factors among nurses in Austria. This study investigates the prevalence, symptomatology, and preferred support measures for SVP among Austrian nurses. **Methods:** A nationwide, cross-sectional, anonymous online survey was conducted September to December 2023 using the SeViD questionnaire (Second Victims in German-speaking Countries), which includes the Big Five Inventory-10 (BFI-10). Statistical analyses included binary logistic regression and multiple linear regression using the bias-corrected and accelerated (BCa) bootstrapping method based on 5000 bootstrap samples. **Results:** A total of 928 participants responded to the questionnaire with a response rate of 15.47%. The participants were on average 42.42 years old and were mainly women (79.63%). Among the respondents, 81.58% (744/912) identified as Second Victims (SVs). The primary cause of becoming an SV was aggressive behavior from patients or relatives. Females reported a higher symptom load than males, and higher agreeableness was linked to increased symptom severity. Notably, 92.47% of SVs who sought help preferred support from colleagues, and the most pronounced desire among participants was to process the event for better understanding. **Conclusions:** The prevalence of SVP among Austrian nurses is alarmingly high, with aggressive behavior identified as a significant trigger. The findings emphasize the necessity for tailored support strategies, including peer support and systematic organizational interventions to mitigate the impact of SVP on nurses and to improve overall patient care. Further research should focus on developing and implementing effective prevention and intervention programs for healthcare professionals in similar settings.

## 1. Introduction

The healthcare sector is fraught with substantial risks, affecting both patients and healthcare practitioners [1]. As societal evolution outpaces the adaptability of healthcare systems, technological advancements exacerbate the industry’s inefficiencies [2,3]. The European Commission’s inquiry into the costs of unsafe care highlights the frequent occurrence of complications among hospitalized patients, many of which are preventable [4]. These complications not only deteriorate patients’ quality of life but also erode trust in healthcare institutions [5].

Therefore, these events harm patients as well as subject healthcare professionals to distress and trauma, possibly making them Second Victims (SVs) [6]. Coined by Albert Wu in 2000, the Second Victim Phenomenon (SVP) initially focused on physicians traumatized by medical mishaps, identifying them as SVs and the patients and their families as First Victims [7]. Scott et al. later expanded this concept to include the trauma experienced by all healthcare workers following adverse events [8].

The European Researcher Network Working on Second Victims (ERNST) recently provided a comprehensive definition of SVs, encompassing any healthcare worker affected by unforeseen adverse patient events or inadvertent errors [9]. The psychological ramifications of such events range from guilt and anxiety to diminished confidence and even contemplation of self-harm. Importantly, the SVP not only affects the well-being of SVs but also undermines the quality of care provided to subsequent patients.

Global efforts have examined the experience of healthcare workers in the context of the SVP [10,11,12,13,14,15], with significant studies conducted in Germany and Austria [16,17,18]. A survey of German nurses revealed a substantial proportion experiencing SVP, often with prolonged recovery periods [19]. Similarly, a recent nationwide survey among Austrian pediatricians uncovered a significant prevalence of self-reported SVs, particularly among outpatient pediatricians [20].

A recent study from Croatia indicates that nurses are more susceptible to adverse events and experience mental health disorders more intensely than doctors [21]. Furthermore, a cross-sectional analysis across Belgian hospitals compared the responses of physicians, midwives, and nurses following adverse events, revealing varied reactions across professions [22]. Nurses, who constitute the largest group among healthcare professionals, play a pivotal role in patient care and often face physical and emotional repercussions following adverse occurrences.

Research suggests that nurses may be reluctant to seek professional support after an incident, due to limited awareness of the SVP and inherent barriers [23,24,25,26]. Organizational support mechanisms are crucial in preventing adverse outcomes, such as exacerbation of harm and employee attrition [27,28].

Although the SVP has been widely studied in other contexts, no research has yet focused on its prevalence among Austrian nurses. The SeViD-A2 study, conducted by the Austrian Second Victim Association and the Wiesbaden Institute for Healthcare Economics and Patient Safety (WiHelP), in collaboration with the Austrian Nurses Association (ÖGKV), seeks to fill this gap by evaluating the prevalence and symptomatology of SVP among Austrian nurses. Additionally, the study also examined preferred support measures as a secondary outcome. The study also seeks to identify demographic, workplace-related, and personality trait determinants associated with the likelihood and severity of SVP. It is well known that an individual’s reaction to stress is influenced not only by personal capacities, such as personality traits and available resources [29,30], but also by the specific characteristics of the stressful situation [31]. Our objective is to gain a deeper understanding of how different factors contribute to stress responses, particularly in the context of adverse events. By analyzing the specific elements and circumstances surrounding these events, we can potentially identify key areas where targeted interventions could be most effective. Such tailored approaches could lead to more effective support mechanisms and improved well-being and resilience among individuals facing similar challenges.

As a first study of this kind, this study aims to investigate the multifaceted nature of the SVP among Austrian nurses, with the goal of informing supportive interventions and organizational protocols to mitigate its adverse consequences.

## 2. Materials and Methods

### 2.1. Design and Conduction of the SeViD-A2 Survey

This nationwide cross-sectional study was conducted among Austrian nurses utilizing the SeViD questionnaire (Second Victims in German-speaking Countries). The detailed development and validation of the questionnaire are outlined in another publication [32] and referenced in the SeViD-I [16], -II [19], and -III [18] as well as the SeViD-A1 [20] publications. The questionnaire was adapted to reflect local conditions, language nuances, and demographic context to ensure its cultural relevance and validity for the Austrian population. Comprising 25 questions across seven dimensions, the questionnaire addressed basic demographics, knowledge and exposure to SVP, the adverse event leading to it, recovery processes, reactions, and support measures. Personality traits, including openness, neuroticism, agreeableness, extraversion, and conscientiousness, were evaluated using the 10-item short version of the Big Five Inventory (BFI-10) [33]. Two additional questions were incorporated, one regarding the association of the adverse event leading to the SVP with the COVID-19 pandemic, and the other inquiring whether those affected feared legal consequences.

### 2.2. Survey Methodology

Employing adaptive questioning, only participants identifying as SVs received questions related to the events leading to the SVP or their symptoms. Respondents were required to answer all questions before progressing to the next item, but they had the flexibility to revise and change their previous responses if necessary. At the end of the survey, participants were given the opportunity to provide comments or suggestions.

### 2.3. Survey Distribution and Data Collection

The survey was distributed via email to members of the Austrian Nurses Association (ÖGKV), totaling approximately 6.000 members. All registered nurses were invited to participate on a voluntary basis, and the survey was distributed exclusively to nursing staff. The invitation letter provided a brief overview of this study’s objectives, an introduction to the SVP, and a link along with a QR code directing participants to the online survey hosted on the SurveyMonkey platform. Data collection was conducted anonymously, with no retention of tokens, cookies, or IP addresses. This study spanned three months, from 15 September to 12 December 2023.

### 2.4. Preparation and Re-Coding of Variables for Statistical Analysis

The procedure for preparing and re-coding variables for statistical analysis followed the same methodology as in the SeViD-A1 study, allowing for direct comparisons. The ordinal-scaled item “SV-status” was dichotomized. Participants who had experienced the SVP once or multiple times were coded as 1, and those who had not as 0. The ordinal-scaled educational status of the nurses was dichotomized as follows: Bachelor, Master, PhD, and Diploma in Healthcare and Nursing (DGKP- ‘Diplom Gesundheits und Krankenpflege’) were assigned a value of 1, while qualified nursing assistants, nursing assistants, and persons without job training were assigned a value of 0. The ordinal-scaled variable working hours was categorized as follows: full-time was assigned a value of 1, and part-time was assigned a value of 0. Nurses with a leading function were assigned a value of 1, and the others were assigned a value of 0 (ordinal-scaled). According to the SeViD-II study, which took place among nursing staff in Germany, we dichotomized nurses working in critical care (e.g., operating theater, intensive care unit, or emergency department) as 1, and all others as 0 (nominal-scaled). To evaluate symptom load, we calculated sum scores for 18 symptoms potentially caused by SVP, based on prior research [19]. Our goal was to capture the overall symptom burden, rather than focusing on individual symptoms. We used positive scoring items for symptom load, with values assigned as follows: “strongly pronounced” = 1, “weakly pronounced” = 0.5, and both “not at all” and “I don’t know” = 0. The variable symptom load was interval-scaled.

Two additional items in relation to symptoms were assessed but excluded from the sum score: “desire for support”, “desire to process the event”. These two items were considered support desires rather than symptoms. Frequency distributions are also, for these as well as all other symptoms, reported in the results section. While symptom load is considered an interval-scaled variable and treated as such in inferential analyses, including multiple linear regression (see Section 2.5), we still report frequency distributions for every symptom pronunciation. By reporting frequencies, even for interval-scaled variables, we aimed to provide insight into the distribution of responses and contextualize the findings beyond the inferential statistics. Recovery time from the critical event was measured on a nominal scale with the following categories: 1 (“less than a day”), 2 (“within a week”), 3 (“within a month”), 4 (“within a year”), 5 (“more than a year”), and 6 (“I have not fully recovered”). It is important to note that this variable was treated as nominal because the response categories do not necessarily reflect a strict ordinal progression. Specifically, the option “I have not fully recovered” does not inherently represent the longest recovery period. Depending on when the critical event occurred, an individual who selected this option could, in theory, have experienced a shorter recovery period than someone who selected an option like “within a year.” For example, a participant who has not yet fully recovered but experienced the event recently may be at an earlier stage of recovery compared to someone who recovered after several months. This lack of a clear temporal hierarchy in the response options justifies treating the variable as nominal rather than ordinal.

### 2.5. Statistical Analysis

The statistical analysis methods also replicated those used in the SeViD-A1 study to ensure comparability.

Descriptive statistics of interval-scaled variables were reported using means (Ms) with standard deviations (SDs) independent of the distributional characteristics [34]. Ordinal variables were summarized using frequencies and percentages for each rank category, while nominal variables were presented by frequencies and percentages for their respective categories. This approach prevents overinterpretation and ensures a clear, comparable presentation of categorical data, especially given the numerous dichotomized variables in the dataset. Since some of them are nominal and others ordinal, using median and quartiles would be inappropriate for both dichotomous ordinal and nominal variables. Thus, we consistently report all ordinal variables using frequencies and percentages. All percentage values refer to the denominator of respondents who answered the specific question. Statistical analysis was performed using SPSS Statistics Version 29 (IBM, New York, NY, USA). A *p*-value lower than 0.05 was considered significant.

To compare the perceived helpfulness of preferred support measures between participants who had encountered the SVP at least once in their professional career and those who had not, we performed a Mann-Whitney U test. The association between predictors such as gender, age, length of professional experience, workplace (acute care vs. others) and the five personality dimensions (openness, neuroticism, agreeableness, extraversion, and conscientiousness) and the dichotomous dependent variable (SVP experienced at least once: yes vs. no) was assessed using binary logistic regression. Bootstrapping with 5000 bootstrap samples (bias-corrected and accelerated (BCa) method was employed to provide more robust confidence intervals. In addition, we performed multiple linear regression with the same predictors and symptom load as a criterion variable. In the next step, we also included the types of adverse events as predictors in a multiple linear regression. Multicollinearity was checked using the bivariate correlation matrix, tolerance, and variance inflation factor (VIF). If detected, predictor variables were mean-centered. The analysis was conducted using the BCa bootstrapping method based on 5000 bootstrapped samples.

Based on previous findings, this study tested the indirect effects of work experience on symptom load via five personality traits using model 4 for parallel mediators from the PROCESS macro for SPSS v4. Bootstrapping with a bias-corrected 95% confidence interval based on 5000 bootstrap samples estimated direct, indirect, and total effects, as shown in Figure 1.

#### Justification for Methods Used: Bootstrapping in Regression Analysis

For linear regression analysis, three key assumptions must be met to ensure confidence in our parameter estimates, confidence intervals (CIs), and standard errors: linearity, normal distribution of residuals, and homoscedasticity. However, these assumptions are often not satisfied in practice [35]. Using the bootstrapping method, we can perform linear regression even when these assumptions have not been met and be confident about our findings. In our analysis, the two latter assumptions were violated, but we proceeded with the analysis as the bootstrapping method allows us to perform linear regression even when such assumptions are violated and obtain robust CIs of parameter estimates and standard errors (see Filed, 2013, for an overview [36]). Furthermore, the bootstrapping method is also routinely applied in Process SPSS Macro by Heyes, 2018. This very widely used macro tests various mediation, moderation, and conditional models using linear regression equations [37].

The assumptions for logistic regressions refer neither to normal distribution nor to variance homogeneity (homoscedasticity). Nevertheless, certain assumptions must be met to perform logistic regressions. They include a part of the independency of observations and the absence of multicollinearity, the linearity in the logit for continuous variables as well as the absence of strongly influential outliers [38]. Validation of these assumptions can lead to inaccurate parameter estimations and standard errors. In our analysis, we did not explicitly test the latter two assumptions—linearity in the logit for continuous variables and the absence of strongly influential outliers—because traditional methods for assessing these assumptions can be somewhat subjective. For instance, determining linearity often relies on visual inspections of scatterplots, which can lead to varying interpretations among analysts. Similarly, identifying influential outliers may depend on arbitrary thresholds, further introducing subjectivity into the analysis. Even though tests for linearity in the logit, such as the Box–Tidwell test, are available, they can be sensitive to sample size, as they evaluate linearity by transforming predictor variables and checking their significance. If the underlying relationship is not linear, the transformation might not adequately capture it. Using transformations to achieve linearity can sometimes lead to overfitting the model to the sample data, especially if too many transformations or interactions are considered. Additionally, it is not robust to outliers and can produce results that are difficult to interpret, see Harrel, 2012 for an overview [39]. Given these challenges, we opted to use bootstrapping, which allows us to derive reliable parameter estimates, confidence intervals, and standard errors even in the presence of violations of these assumptions [40,41]. In addition, we conducted ROC (Receiver Operating Characteristic) statistics to determine the area under the curve (AUC) and assess its asymptotic significance. An AUC above 0.6 would be considered as an acceptable discrimination [42]. This analysis provides insight into the model’s ability to discriminate between the binary outcomes, helping us evaluate its predictive performance.

We employed 5000 bootstrap samples for our analysis. To confirm the adequacy of this sample size, we conducted the bootstrap procedure multiple times and observed the consistency of the estimates. Our findings indicated that the estimates remained stable across runs, suggesting that the number of samples is sufficient. Although Wilcox recommends a minimum of 599 bootstrap samples, this guideline served as a starting point for our analysis [43]. Given our computational resources, we opted to use a larger sample size to ensure even more stable estimates.

## 3. Results

### 3.1. Descriptive Baseline Analysis

Out of the 6000 participants addressed, 928 responded to the questionnaire, which corresponds to a response rate of 15.47%. Among these respondents, 82% successfully completed the entire survey. The mean time for participation was 9 min and 25 s. Table 1 shows the participants baseline characteristics.

Most participants were female (79.63%), 20.04% were male, and 0.3% stated having a diverse (non-binary) gender identity. The mean age of the participants was 42.42 years (SD 10.36; range 18–65) and the mean work experience was 18.64 years (SD 11.50; range 1–45). Most participants worked full-time (61.96%) and in shifts (70.48%).

Regarding their acquaintance with the term “Second Victim”, 67.54% (616) of the participants indicated not having heard of it prior to the survey. After a brief explanation, 81.58% (744) stated that they now recognize themselves as SVs. A substantial majority (63.38%) reported being an SV on multiple occasions, as shown in Figure 2. Of the responding participants, 18.42% indicated that they did not identify as SVs. Furthermore, 59.36% (425) mentioned that they were SVs within the past twelve months.

Most participants identified aggressive behavior of patients or their relatives (37.43%) as the primary event leading to SVP, followed by the unexpected death or suicide of a patient (24.02%), as shown in Table 2. Notably, 18.58% of the key incidents leading to SVP were related to the COVID-19 pandemic.

Some participants specified their key events further (similar answers were summarized):

“Medication mixed up, patient sedated”

“Missed care”

“Lifeless child at birth”

“Bullying from colleagues”

“Reports of domestic violence”

“Aggressive behavior by doctors”

“Severely injured patient”

“Stillbirths”

“Death of young patients”

“Lack of patient care with fatal outcome”

“Serious injury to a young patient due to external violence”

“Suicide attempt by a patient”

“Unjustified accusations by a relative”

“Postpartum hemorrhage”

“Colleague who died on her own ward as a patient on my shift”

“Aggressive behavior of a nurse towards a patient”

“Sexual harassment by patients”

“Number of deaths during COVID”

“Verbally aggressive behavior by middle managers”

“Expected death of a very young patient”

“Patient bled to death within 15 min”

“Dismissal of a colleague”

“Moral stress”

Over half of the participants (56.56%, 405) received help after a traumatizing adverse event; 30.73% (220) did not receive any help but had not asked for it, while 12.71% (91) stated they were denied help even though they actively asked for it. The vast majority received help from their colleagues, as shown in Table 3.

The recovery time of the SVs varied: 29.91% (198) recovered within one month, 27.95% (185) within one week, and 12.99% (86) stated that they had not yet fully recovered. In relation to their reactions to the adverse event, participants were presented with possible symptoms and asked to rate them according to how severe they were. The most pronounced symptoms were insomnia or excessive need for sleep, reliving the situation in similar professional situations and psychosomatic reactions like head- or backaches as shown in Table 4. The desire to receive support from others was strongly pronounced in 39.43% (261) of participants. However, the most expressed feeling was the desire to process the event for better understanding, it being strongly pronounced in 46.22% (306) of the respondents.

When asked about the fear of legal consequences, the majority (83.83%, 555) of respondents stated that it was weakly or not pronounced at all. Only 14.5% (96) stated that the fear of legal consequences was strongly pronounced.

Participants were also asked to rate possible supporting measures. Support measures were scored on a five-point descending Likert scale ranging from one, “very helpful”, to four, “not helpful at all” (“not helpful at all” = 4, “rather not helpful” = 3, “rather helpful” = 2 and “very helpful” = 1) are shown in Table 5. The option “I cannot judge this” was treated as a missing value.

Overall, support measures are viewed positively, indicating moderate to high helpfulness. Non-SVs find professional counseling more helpful than SVs, while SVs place greater importance on discussing emotional and ethical thoughts and receiving formal emotional support. SVs also prefer quick crisis intervention, finding it more helpful than non-SVs. Both groups equally value clear and timely information, support for work continuation, and effective communication with patients and relatives. SVs consider guidelines during serious events and secure reporting for future prevention more helpful, whereas non-SVs rate legal counseling as more beneficial.

### 3.2. Factors Associated with the Likelihood of Becoming a Second Victim

The results of the binary logistic regression analysis showed that being female was associated with a higher likelihood of becoming an SV (regression Coefficient B = −0.53, BCa 95% CI [−0.99, −0.13]). The odds ratio (OR) was 0.59, with a 95% CI [0.38, 0.91]. Other healthcare worker-centered and work environment-related predictors were not correlated with a higher or lower chance of becoming an SV as shown in Table 6 (Nagelkerke Pseudo-R^2^ = 0.03; AUC = 0.6, *p* < 0.001).

### 3.3. Factors Associated with the Symptom Load After the SV Experience

We conducted a multiple linear regression analysis to examine the correlation between demographic factors, healthcare worker-related characteristics, workplace-related factors, and personality traits and the symptom load experienced after an SV experience. The outcome variable was the sum and severity of symptoms suffered following an SV experience (Table 7).

The analysis indicates that among the variables tested, gender and agreeableness were significantly correlated to the symptom load experienced after an SV experience. Females reported a higher symptom load compared to males, whereas higher agreeableness was linked to a higher symptom load. Other factors, including age, professional experience, education, job status, management role, specific workplace, openness, extraversion, and neuroticism, were not significant predictors of symptom load.

In the next step, we included the types of adverse events in the regression equation as shown in Table 8.

The results show that no demographic, workplace-related, or personality trait factor was significant in predicting symptom load after including the types of adverse events in the regression equation. Interestingly, all types of incidents involving patient harm, unexpected death or suicide of a patient, unexpected death or suicide of a colleague, and aggressive behavior by patients or relatives were more strongly correlated with higher symptom load than events without patient harm.

### 3.4. Testing the Mediational Model

Based on previously published studies, we tested a mediational model with personality traits as a mediator in the relationship between years of professional experience and symptom load after the SV experience using SPSS Process macro, Model 4 for parallel mediators.

The results revealed neither significant direct (unstandardized regression coefficient (B = −0.004, bootstrapped 95% CI [−0.03, 0.002]) nor indirect effects of professional experience on symptom load (see Table 9). The total effect was also not significant (B = −0.008 bootstrapped 95% CI [−0.04, 0.002]). The whole model with predictor and mediator predicted 11% of the symptom load variance, F(6,756) = 1.64, *p* = 0.13.

## 4. Discussion

The present study reveals several critical insights into the SVP among Austrian nurses. A striking 82% of the nurses surveyed identified themselves as SVs, underscoring the pervasive nature of SVP in this group. The most common type of event leading to SVP was aggressive behavior from patients or relatives, reported by 37.43% of participants. In terms of symptoms, insomnia or excessive need for sleep, reliving the situation in similar professional contexts, and psychosomatic reactions such as headaches and backaches were the most frequently reported. The desire to process the event for better understanding and the need for support from others were also strongly pronounced among respondents. Colleagues emerged as the most favored source of support, with 92.47% of participants seeking help from their peers, highlighting the critical role of peer support in coping with SVP.

The notable decrease in participants over the age of 60 compared to SeViD-A1 [20], despite similar levels of experience, introduces a critical demographic shift. This younger sample, with higher neuroticism and symptom load, suggests that age and personality traits play a significant role in how healthcare professionals experience and cope with serious events. Younger professionals, possibly less experienced in managing high-stress situations, may exhibit heightened vulnerability, leading to more pronounced emotional and psychological symptoms [44]. This demographic shift underscores the need for tailored interventions that consider the unique stress responses and coping mechanisms of younger healthcare workers.

This study’s finding of no significant correlation between most personality traits and SV status, except for agreeableness in relation to event type, points to a complex interaction between individual differences and the impact of serious events. The restricted variance in personality traits within the sample might obscure potential relationships, suggesting that personality may influence the SVP in more nuanced ways than previously understood.

Nevertheless, the emergence of agreeableness as significant only after accounting for the types of adverse events highlights its context-dependent role in SV development. This suggests that agreeableness does not universally predispose individuals to SVP; rather, it becomes relevant in specific situations involving interpersonal dynamics or emotional sensitivity [45]. Typically associated with empathy and cooperation, agreeableness may lead to greater emotional distress in events involving patient harm, where an individual’s desire for harmony is challenged [45]. The finding that agreeableness was not significant anymore after event types were considered implies that it may interact with the nature of adverse events to indirectly influence SV outcomes. This emphasizes the necessity of considering both personality traits and specific event circumstances when predicting SVP. To summarize, the relationship between agreeableness and event type was noteworthy, but this finding should be interpreted cautiously given the restricted range of personality scores, which could limit the generalizability of the results.

Interestingly, this study’s higher observed levels of neuroticism among nurses may be influenced by recent research indicating that younger professionals report higher neuroticism compared to older cohorts, potentially affecting their responses to adverse events [46].

The persistent finding that women are at a higher risk of becoming SVs aligns with some previous research [16] but stays in contrast to a previous survey among German nurses [19]. It raises questions about the underlying causes. The gender disparity in SVP risk may be influenced by specific work environments and roles predominantly occupied by men and women, suggesting that job-specific factors and gender-specific personality traits both play significant roles in susceptibility to SVP [47,48]. Gender-specific factors, such as differences in emotional labor and coping mechanisms may contribute to the observed disparity, but further research is needed to fully understand the extent and nature of these influences.

This study’s failure to identify significant mediation effects could be attributed to restricted data variance due to the younger sample. In our analysis of the SeViD-III study, we found that years of job experience were no longer significant predictors of symptom load following a SV event after accounting for neuroticism in the multiple linear regression. This led us to conduct an inductive, bottom-up mediation analysis, which revealed a negative indirect effect of years of experience on symptom load through neuroticism. Similar findings were noted in the SeViD-IX and SeViD-A1 studies. We interpreted this negative indirect effect as potentially indicative of a self-selection. Specifically, individuals with higher neuroticism may have opted to leave their positions or change careers, while those with lower neuroticism remained, resulting in lower neuroticism scores among the more experienced workers. This could also mean that more experienced employees benefit from stronger social networks and access to emotional support at work, further mitigating their symptom load. Additionally, recent research into the baby boomer generation suggests that neuroticism tends to decline with age [46]. Since age correlates with years of experience, we posited that this decline could lead to lower neuroticism and, consequently, reduced symptom load. However, our study found no direct correlation between neuroticism and symptom load, which hindered the manifestation of a significant indirect effect. Although the theoretical framework is sound and neuroticism serves as a mediating variable in various contexts, including the Job Demands–Resources model [29,49], the specific characteristics of our sample may not have provided enough variability in neuroticism to manifest a clear mediation effect. In addition, although we explored all Big Five personality traits, none emerged as significant predictors of symptom load. While certain traits, like neuroticism and agreeableness, are frequently discussed in the literature as being influential in the context of stress and emotional well-being, our findings suggest that their predictive power may depend on specific circumstances or additional moderating factors. This appears to be the case for agreeableness, which lost its significance once the type of adverse event was included in the equation as previously noted. In light of these findings, we recommend that future research includes a longitudinal design and broader demographic, particularly older healthcare workers, and a collection of further contextual variables to explore how age and experience correlate with personality traits and symptom load.

Moreover, the type of event seems to play a significant role as a primary determinant of SVs, highlighting the importance of event severity and nature in shaping the impact on healthcare workers. Violence against nurses is a well-documented issue, with significant negative impacts on their personal and professional well-being. This includes verbal abuse, physical violence, and sexual harassment, leading to increased job stress, low morale, and higher turnover rates [50,51,52,53,54]. The high prevalence of workplace violence in Austria could partly explain the elevated rate of SVP among nurses in this study. This finding underscores the need for robust systems to address various types of events, particularly through targeted prevention strategies like de-escalation training and communication [55,56].

In the current study, we emphasized workplace violence as a significant factor contributing to the SVP. However, research shows that systemic issues in healthcare, such as staffing shortages, high patient loads, and the stressful nature of nursing work, are significant contributors to the SVP. These conditions create chronic stress for healthcare workers, increasing burnout and reducing their ability to deliver quality care. For example, inadequate nurse-to-patient ratios have been directly linked to adverse patient outcomes and higher rates of nurse burnout [57], both of which can potentially intensify SVP effects [24]. Although this study focused on workplace violence as a key driver of SVP, future research should consider these broader systemic factors. We did not collect data on these variables but incorporating them in future studies could provide a more comprehensive understanding of the SVP in healthcare environments. Additionally, while we reported an 82% prevalence of SVP, it is important to interpret this figure with caution, given the potential for response and social desirability biases.

This study reveals a divergence in support needs, with SVs prioritizing emotional support and non-SVs valuing legal consultation. If healthcare workers do not identify themselves as SVs, they may even still exhibit symptoms after an adverse event as first demonstrated in the SeViD-IX study among German general practitioners [58]. Implementing support strategies that address these specific needs can enhance the overall effectiveness of interventions and ensure that all affected healthcare professionals receive appropriate and comprehensive support.

The higher prevalence of SVP among Austrian nurses (82%) may be influenced by missed nursing care (MNC), which is defined as any required patient care that is omitted or delayed [59]. The MISS-Care Austria 2022 study found that 84.4% of nurses reported omitting at least one nursing intervention in the preceding two weeks. Factors contributing to MNC include multitasking, staff shortages, and personal exhaustion. This high rate of MNC could lead to SVP, particularly in environments where mistakes are made due to resource limitations [60].

Previous studies suggest that nurses may make more external attributions following serious errors, potentially hindering constructive responses and learning from mistakes. This tendency may be linked to the strong professional ethos among nurses, which emphasizes personal responsibility. Addressing this issue requires careful management to encourage reflective practice and positive behavior change following errors [61].

Regardless of the type or severity of the adverse event, nurses report significant physical and emotional manifestations, underscoring the need for appropriate support. However, many do not proactively seek help, possibly due to limited awareness of SVP or perceived barriers [24]. Support from colleagues is often the most appreciated, but systematic organizational interventions are necessary to ensure all healthcare workers receive the support they need [22].

Our study adds to the existing literature on the SVP by confirming the anticipated high prevalence of SVP among healthcare professionals [62,63,64]. However, the unexpected lack of a direct correlation between personality traits—specifically neuroticism as measured by the BFI-10—and SVP symptoms challenges previous assumptions regarding the role of individual personality factors in predicting SVP experiences. Moreover, the mediation analysis did not yield the anticipated results, suggesting that the sampling method and the included population, such as young healthcare workers, can influence the outcomes of the mediation analysis and the associations between personality traits, SVP, and symptom load following an adverse event. According to other SeViD studies, workplace characteristics and specific predictors related to the working environment did not demonstrate significant effects; the exception was that working in the outpatient setting emerged as a notable predictor in our SeViD-A1 and SeViD-IX studies. Furthermore, for the first time, our study incorporated the type of critical incident experienced as a predictor, revealing its importance in understanding SVP. Future research should further investigate the impact of incident types on the SVP experience, emphasizing the complexity of the factors that contribute to this phenomenon. In summary, while our findings align with previous research regarding the prevalence of SVP, they also introduce new insights into the predictors of SVP, warranting further exploration in subsequent studies.

While this study provides valuable insights into the SVP among Austrian nurses, several limitations should be acknowledged. First, the response rate of 15.47%, although reasonable for voluntary surveys, raises concerns about response bias. It is possible that those who participated in the survey were more likely to have experienced or been affected by SVP, potentially leading to an overestimation of the prevalence of SVP among Austrian nurses. Conversely, those who did not participate might have different experiences or less severe symptoms, which could mean that the findings do not fully represent the entire nursing population.

Another potential limitation is social desirability bias, which may have influenced how participants responded to the survey questions. Nurses may have felt compelled to provide answers they perceived as socially acceptable or aligned with the expectations of their profession, particularly in questions related to admitting mistakes or discussing the severity of their symptoms. This bias could result in an underreporting of more sensitive issues, such as the impact of SVP on their mental health or the extent of missed nursing care due to resource constraints.

The cross-sectional design of this study limits the ability to draw causal inferences. While associations between factors like personality traits, event types, and SVP symptomatology were observed, it is challenging to determine the direction of these relationships or to account for potential confounding variables. Longitudinal studies would be needed to better understand the temporal dynamics of SVP and the long-term effects on healthcare professionals.

The reliance on self-reported data also introduces potential inaccuracies due to recall bias. Participants may have difficulties with accurately remembering the details of past events or their reactions to them, particularly when these events occurred a long time ago. However, it is important to note that the key incidents reported, such as aggressive behavior or unexpected patient deaths, were often drastic and highly traumatic. Such events are likely to leave a strong impression, potentially alleviating some of the concerns about recall bias, as individuals are more likely to vividly remember and accurately report severe and emotionally charged experiences.

In light of these limitations, the findings of this study should be interpreted with caution. Future research could address these issues by incorporating longitudinal designs where possible to better capture the complex dynamics of SVP among healthcare workers.

However, it is important to note that the results of this study are consistent with numerous other studies within the SeViD study program and the broader body of research on the SVP. These studies have consistently demonstrated a high prevalence of SVP across various healthcare professions and settings, as well as a strong need and desire for peer support programs. The robust findings from these studies further support the conclusions of our research, underscoring the critical importance of implementing effective support systems to address the pervasive impact of the SVP in healthcare.

Future research should develop targeted interventions like peer support programs and assess their long-term effectiveness in helping nurses recover from SVP. Cross-national studies can identify if SVP causes vary across healthcare systems, while prevention-focused research should explore proactive strategies, such as training and policy changes. These efforts will help reduce SVP incidence and improve nurses’ resilience in high-stress environments.

## 5. Conclusions

The findings from this study offer critical insights into the prevalence, causes, and consequences of the SVP among Austrian nurses, highlighting the need for targeted interventions and systemic changes within healthcare organizations. With 82% of the nurses surveyed identifying as SVs, it is evident that the SVP is a pervasive issue in nursing, exacerbated by both the nature of the work environment and the specific stressors encountered by nurses.

This study underscores the urgent need for healthcare organizations to recognize and address the SVP among nurses. By implementing comprehensive support systems, improving workplace safety, and addressing the systemic issues that contribute to SVP, healthcare institutions can not only improve the well-being of their nursing staff but also enhance the overall quality of patient care. The findings of this study should serve as a call to action for healthcare leaders to prioritize the mental and emotional health of their workers, recognizing that the well-being of nurses is integral to the success of the healthcare system as a whole.

## Figures and Tables

**Figure 1 healthcare-12-02061-f001:**
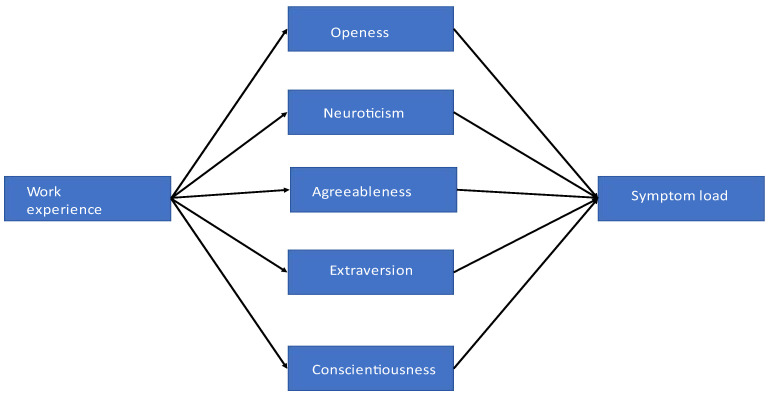
Parallel-mediation model. Work experience: length of professional experience in years. Openness, neuroticism, agreeableness, extraversion, and conscientiousness: Big Five personality traits. Symptom load: the sum of symptoms after the SVP experience. Adapted from SeViD-A1 Study [20].

**Figure 2 healthcare-12-02061-f002:**
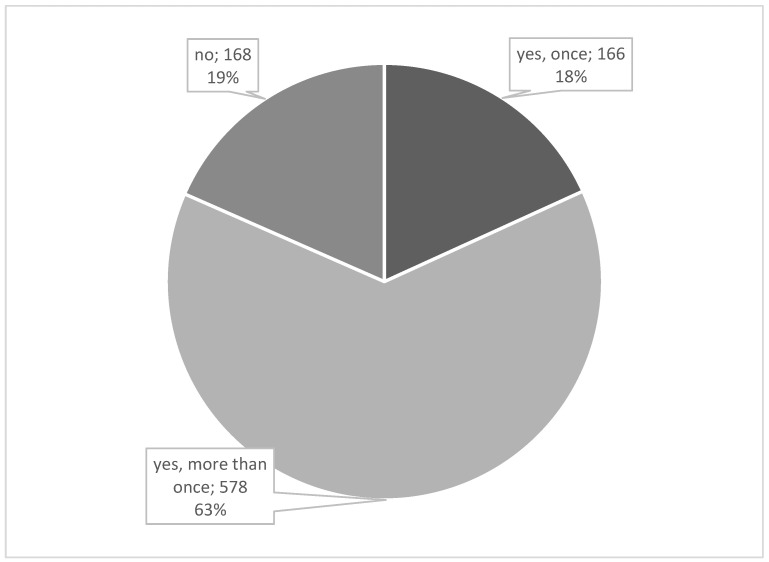
Have you ever experienced the SVP yourself? n = 912.

**Table 1 healthcare-12-02061-t001:** Baseline characteristics of the participants. n = 928.

Characteristics		Percentage of Participants (n)
Gender	Female	79.63 (739)
	Male	20.04 (186)
	Diverse	0.3 (3)
Age	18–30 years	15.31 (142)
	31–40 years	27.15 (252)
	41–50 years	31.04 (288)
	51–60 years	24.36 (226)
	>60 years	2.15 (20)
Work experience	1–10 years	31.48 (292)
	11–20 years	26.4 (245)
	21–30 years	24.45 (227)
	>30 years	17.69 (164)
Specialization	None	21.88 (203)
	Management tasks	22.95 (213)
	Intensive care	20.04 (186)
	Other	74.34 (690)
Education	Master	16.16 (150)
	Bachelor	21.34 (198)
	PhD	0.97 (9)
	Nursing Diploma	83.30 (773)
	Nursing Specialist Assistant	3.32 (30)
	Nursing Assistant	6.25 (58)
	none	0.86 (8)

**Table 2 healthcare-12-02061-t002:** Types of the most formative adverse event (key experiences). n = 716.

Type of Event	Percentage of Participants (n)
Incident with patient harm	14.25 (102)
Incident without patient harm (near miss)	14.11 (101)
Unexpected death/suicide of a patient	24.02 (172)
Unexpected death/suicide of a colleague	5.03 (36)
Aggressive behavior of patients or relatives	37.43 (268)
Other	5.17 (37)

**Table 3 healthcare-12-02061-t003:** (Occupational) Groups which those affected by SVP sought out for help (selection of multiple sources possible). n = 372.

(Occupational) Group	Percentage of Participants (n)
Colleagues	92.47 (344)
Supervisors	38.44 (143)
Management	5.38 (20)
Family/Friends	34.68 (129)
Counsellors/Psychotherapists/Psychologic Counseling	16.13 (60)

**Table 4 healthcare-12-02061-t004:** Frequencies of severeness of different symptoms experienced after SV event. n = 662.

Symptom	Not at All	Weakly Pronounced	Strongly Pronounced	I Don’t Know
Fear of exclusion by colleagues	407 (61.48%)	145 (21.90%)	90 (13.6%)	20 (3.02%)
Fear of losing the job	475 (71.75%)	108 (16.31%)	64 (9.67%)	15 (2.27%)
Listlessness	265 (40.03%)	258 (38.97%)	124 (18.73%)	15 (2.27%)
Depressive mood	195 (29.46%)	314 (47.43%)	140 (21.15%)	13 (1.96%)
Concentration difficulties	198 (29.91%)	304 (45.92%)	144 (21.75%)	16 (2.42%)
Reliving the situation outside of professional life	271 (40.94%)	220 (33.23%)	142 (21.45%)	29 (4.38%)
Reliving the situation in similar professional situations	127 (19.18%)	290 (43.81%)	227 (34.29%)	18 (2.72%)
Aggressive, risky behavior	510 (77.04%)	94 (14.20%)	29 (4.38%)	29 (4.38%)
Defensive, overly cautious behavior	220 (33.23%)	267 (40.33%)	160 (24.17%)	15 (2.27%)
Psychosomatic reactions (head- or backaches)	195 (29.46%)	204 (30.82%)	225 (33.99%)	38 (5.74%)
Insomnia or excessive need for sleep	119 (17.98%)	234 (35.35%)	298 (45.02%)	11 (1.66%)
Use of alcohol/drugs because of event	490 (74.02%)	135 (20.39%)	26 (3.93%)	11 (1.66%)
Feelings of shame	408 (61.63%)	154 (23.26%)	85 (12.84%)	15 (2.27%)
Feelings of guilt	260 (39.27%)	235 (35.50%)	153 (23.11%)	14 (2.11%)
Self-doubts	180 (27.19%)	271 (40.94%)	202 (30.51%)	9 (1.36%)
Social isolation	433 (65.41%)	152 (22.96%)	67 (10.12%)	10 (1.51%)
Anger towards others	254 (38.37%)	220 (33.23%)	178 (26.89%)	10 (1.51%)
Anger towards myself	365 (55.14%)	180 (27.19%)	103 (15.56%)	14 (2.11%)
Desire for support from others	125 (18.88%)	251 (37.92%)	261 (39.43%)	25 (3.78%)
Desire to process the event for better understanding	109 (16.47%)	221 (33.38%)	306 (46.22%)	26 (3.93%)

**Table 5 healthcare-12-02061-t005:** Differences in the rating of potential support measures following an adverse event between participants who reported prior experience with SVP and those who reported no such experience in their careers. n = 775.

Support Measure	M	SD	M1	SD1	M2	SD2	*p*
	All Participants		SVs		Non-SVs		
The possibility to take time off from work directly to process the event	1.70	0.85	1.72	0.86	1.63	0.82	0.93
Access to professional counseling or psychological/psychiatric consultations (crisis intervention)	1.64	0.96	1.65	0.96	1.58	0.97	<0.01
The possibility to discuss my emotional/ethical thoughts	1.57	0.87	1.53	0.84	1.75	0.98	<0.01
Clear and timely information regarding the course of action after a serious event (e.g., damage analysis, error report)	1.55	0.87	1.54	0.86	1.59	0.90	0.02
Formal emotional support in the sense of organized collegial help	1.65	0.92	1.62	0.91	1.79	0.98	0.01
Informal emotional support	1.75	0.98	1.75	1.00	1.79	0.89	0.01
Quick processing of the situation/quick crisis intervention (in a team or individually)	1.43	0.81	1.41	0.78	1.53	0.91	0.01
Support/Mentoring when continuing to work with patients	2.01	1.05	1.99	1.04	2.10	1.11	0.6
Support when communicating with patients and/or relatives	1.97	1.06	1.97	1.04	1.99	1.13	0.5
Guidelines regarding the role/activities expected of me during a serious event	1.94	1.05	1.91	1.04	2.06	1.09	0.46
Support to be able to take an active role in the processing of the event	1.67	0.89	1.66	0.90	1.71	0.87	0.02
A secure possibility to give information on how to prevent similar events in the future	1.58	0.90	1.55	0.87	1.72	1.01	0.04
The possibility to access legal consultation after a severe event	1.70	0.85	1.72	0.86	1.63	0.82	0.01

SV: Second Victim; SVP: Second Victim Phenomenon; M and SD: mean and standard deviation of the participants who completed the survey (n = 353) regardless of the SV status; M1 and SD1: mean and standard deviation of the participants who reported that they have already experienced SVP; M2 and SD2: the group of participants who reported that they have not experienced SVP in their career; *p*: a *p*-value of the Mann–Whitney U-test comparing the rating of the support measure between the group of participants who reported that they have already experienced SVP and the group of participants that reported that they have not experienced SVP in their career, support measures were scored on a four-point descending Likert scale (“not helpful at all” = 4 to “very helpful” = 1).

**Table 6 healthcare-12-02061-t006:** Factors associated with the likelihood of becoming an SV. Results of binary-logistic regression. n = 763.

Predictor	Regression Coefficient Bwith BCa 95% CI	*p*	Odds Ratio Exp(B) ^1^	Odds Ratio 95% CI	
				Lower	Upper
Gender ^2^	−0.53 [−0.99, −0.13]	0.03	0.59	0.38	0.91
Age	−0.01 [−0.40,0.03]	0.64	0.99	0.96	1.03
Professional experience (years)	0.01 [−0.25, 1.22]	0.59	1.01	0.98	1.04
Education ^3^	0.49 [−0.25, 0.65]	0.13	1.62	0.83	3.16
Full-time or part-time job ^4^	0.20 [−0.33, 0.70]	0.31	1.22	0.81	1.83
Leadership ^5^	0.17 [−0.40, 0.47]	0.61	1.19	0.65	2.18
Workplace ^6^	0.02 [−0.12, 0.41]	0.93	1.02	0.69	1.52
Openness	0.16 [−0.18, 0.54]	0.20	1.17	0.90	1.52
Conscientiousness	0.14 [−0.51, 0.16]	0.38	1.16	0.83	1.61
Extraversion	−0.20 [−0.05, 0.42]	0.25	0.82	0.59	1.13
Agreeableness	−0.19 [−0.46, 0.15]	0.27	1.18	0.92	1.52
Neuroticism	1.16 [−0.52, 3.18]	0.17	0.83	0.61	1.12

Outcome is experienced SV status (dichotomous yes 1 vs. no 0); ^1^ exponentiation of the B Coefficient; ^2^ referent category is male; ^3^ referent category bachelor, master, PhD, diploma; ^4^ referent category is part time job; ^5^ referent category is having a leadership role; ^6^ referent category is acute care, intensive station unit or operation theater; BCa 95% CI: bias-corrected and accelerated bootstrapping 95% confidence intervals based on 5000 bootstrap samples. Please note that we excluded three individuals from the analysis due to their diverse gender identities (i.e., not male or female) and their small number within this category. This exclusion was made to maintain the integrity and clarity of the analysis.

**Table 7 healthcare-12-02061-t007:** Demographic, workplace-related, and personality trait factors associated with the symptom load after the SV experience. Results of multiple linear regression, n = 763, (R^2^ = 0.04); F(12, 450) = 1.75, *p* = 0.05.

Predictor	Unstandardized Regression Coefficient B	*p*	BCa 95% CI	
			Lower	Upper
Gender ^1^	−0.91	0.03	−1.68	−0.07
Age	−0.02	0.48	−0.08	0.04
Professional experience (years)	0.01	0.40	−0.05	0.06
Education ^2^	0.41	0.62	−0.78	1.67
Full-time or part-time job ^3^	0.28	0.61	−0.36	0.92
Management role ^4^	0.23	0.12	−0.60	1.20
Workplace ^5^	−0.17	0.90	−0.75	0.45
Openness	0.31	0.73	−0.07	0.71
Conscientiousness	0.03	0.93	−0.46	0.59
Extraversion	0.11	0.82	−0.54	0.74
Agreeableness	0.49	0.02	0.09	0.90
Neuroticism	0.06	0.48	−0.39	0.52

Outcome is the sum and severity of suffered symptoms after an SV experience; ^1^ referent category is male; ^2^ referent category bachelor, master, PhD, diploma; ^3^ referent category is part time job; ^4^ referent category is having a leadership role; ^5^ referent category is acute care, intensive station unit or operation theater; Lower BCa 95% CI and Upper BCa 95% CI: lower and upper limits of 95% bias-corrected and accelerated bootstrapped confidence interval of unstandardized regression coefficient B based on 5000 bootstrap samples. Please note that we excluded three individuals from the analysis due to their diverse gender identities (i.e., not male or female) and their small number within this category. This exclusion was made to maintain the integrity and clarity of the analysis.

**Table 8 healthcare-12-02061-t008:** Demographic, workplace-related, and personality trait factors, along with the type of adverse event associated with the symptom load after the SV experience. Results of multiple linear regression analysis are presented, n = 763, R^2^ = 0.16, F(16,746) = 8.94, *p* < 0.001.

Predictor	Unstandardized Regression Coefficient B	*p*	BCa 95% CI	
			Lower	Upper
Gender ^1^	0.46	0.21	−1.19	0.29
Age	0.02	0.54	−0.07	0.04
Professional experience (years)	0.004	0.86	−0.04	0.05
Education ^2^	0.26	0.60	−0.67	1.18
Full-time or part-time job ^3^	0.26	0.40	−0.35	0.83
Management role ^4^	0.05	0.91	−0.91	0.82
Workplace ^5^	0.10	0.75	−0.74	0.46
Openness	0.18	0.30	−0.17	0.55
Conscientiousness	0.08	0.76	−0.35	0.55
Extraversion	0.19	0.47	−0.33	0.68
Agreeableness	0.47	0.02	0.08	0.87
Neuroticism	0.31	0.17	−0.14	0.78
Incident with patient harm ^6^	4.49	<0.001	3.65	5.40
Unexpected death/suicide of a patient ^6^	3.12	<0.001	2.33	3.92
Unexpected death/suicide of a colleague ^6^	2.39	<0.001	1.10	3.75
Aggressive behavior of patients or relatives ^6^	3.31	<0.001	2.60	3.99

Outcome is the sum and severity of suffered symptoms after an SV experience; ^1^ referent category is male; ^2^ referent category bachelor, master, PhD, diploma; ^3^ referent category is part time job; ^4^ referent category is having a leadership role; ^5^ referent category is acute care, intensive station unit or operation theater; ^6^ referent category is incident without a patient harm; lower BCa 95% CI and Upper BCa 95% CI: lower and upper limits of 95% bias-corrected and accelerated bootstrapped confidence interval of unstandardized regression coefficient B based on 5000 bootstrap samples. Please note that we excluded three individuals from the analysis due to their diverse gender identities (i.e., not male or female) and their small number within this category. This exclusion was made to maintain the integrity and clarity of the analysis.

**Table 9 healthcare-12-02061-t009:** Unstandardized indirect effects of the length of professional experience as a nurse in years on symptom load caused by SV experience via the Big Five personality traits.

	Unstandardized Effect	BootLLCI	BootULCI
Total unstandardized indirect effect	−0.004	−0.001	0.002
Openness	−0.001	−0.004	0.0004
Conscientiousness	−0.0001	−0.003	0.002
Extraversion	−0.001	−0.003	0.002
Agreeableness	−0.002	−0.005	0.001
Neuroticism	−0.0004	−0.005	0.004

BootLLCI, BootULCI: lower and upper limits of 95% confidence interval based on 5000 bias-corrected bootstrapped samples.

## Data Availability

The data presented in this study are available on request from the corresponding author.

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
