# Peer review of "Second Victims Among Austrian Nurses (SeViD-A2 Study)"

_healthcare, 2024, doi:10.3390/healthcare12202061_

Round 1

Reviewer 1 Report

Comments and Suggestions for Authors

Dear authors

The study addresses an important issue: the Second Victim Phenomenon (SVP) among nurses in Austria. The research appears to be structured appropriately for cross-sectional analysis, clearly focusing on identifying prevalence, symptomatology, and preferred support measures. However, some aspects of the text could be refined to enhance scientific rigour.

Introduction:

The introduction could more explicitly articulate the knowledge gap. While it mentions studies in Germany and Austria, it does not clearly explain what is still unknown about SVP among Austrian nurses or why this study is essential.

Moreover, while it references the history of SVP and some existing studies, the introduction could benefit from more recent and broader literature citations highlighting the current state of SVP research in other countries, mainly focusing on nurses.

Materials and Methods section

The description of the methodology lacks sufficient detail. There is no clear explanation of how the adaptive questionnaire was designed, nor is there information on its validation specifically for the Austrian context. You provide more details on developing and validating the questionnaire for the Austrian population, even if the instrument was adapted from previous versions. Explain how the questions were culturally relevant and valid for this population. Additionally, it would be helpful to describe the validation process of the adaptive questionnaire.

There is no mention of whether reminders were sent to improve the response rate, nor is it clear how many responses were considered valid from the 6,000 invitations. Moreover, there is no mention of inclusion or exclusion criteria for participating in the study. This raises concerns about whether the sample is representative of the target population. You specify the inclusion and exclusion criteria (if any) for participants. For example, was a minimum number of years of nursing experience required? Are we only those in active practice included?

In the statistical analysis section, you should clarify the distinction between ordinal and nominal variables and ensure they are correctly analysed and reported.

The binary logistic regression includes several predictors (e.g., gender, age, personality traits). Still, the relationship between these predictors and the dependent variable (SV experienced at least once: yes vs. no) may not be linear or direct.

You should explain more precisely why bootstrapping was chosen to address any assumptions not met by standard parametric tests (e.g., non-normality of residuals, small sample sizes). Include a rationale for using 5000 bootstrap samples and how this choice affects the stability f the estimates.

You should ensure that the assumptions of binary logistic regression are met, such as the linearity of the logit for continuous predictors like age. Additionally, more details on the model diagnostics, such as model fit statistics (e.g., AIC, BIC) and pseudo-R-squared values, should be included to evaluate how well the model explains the variation in the data.

Discussion: The discussion mentions that personality traits had no significant correlation with SV status, except for agreeableness about event type. This is a surprising result but is not discussed in depth. The discussion should also reflect the restricted variance in personality traits mentioned earlier.

The gender disparity finding (that women are more likely to be Second Victims) is mentioned, but the explanation provided (work environment or gender-specific traits) is relatively vague.

It would be best to clarify how gender-specific factors may contribute to this disparity—more concrete suggestions.

The failure to identify significant mediation effects is attributed to restricted variance in the data. However, this brief explanation does not fully address potential methodological or theoretical reasons for the lack of mediation effects.

You should provide a more in-depth discussion of why the mediation analysis might have failed. Consider discussing whether the theoretical model was appropriately specified or if there were methodological challenges (e.g., insufficient power). Suggest how future studies could be designed to test better mediation effects (e.g., larger sample sizes, different measures).

The discussion highlights workplace violence as a primary driver of SVP. Still, it lacks a critical analysis of how systemic issues in the healthcare system may contribute to this phenomenon beyond violence. It would be best if you expanded the discussion to systemic problems in healthcare, such as staffing shortages, high patient loads, and the stressful nature of nursing work. These factors likely contribute to the high prevalence of SVP and should be addressed alongside the discussion on workplace violence.The conclusion draws strong generalisations about the pervasiveness of SVP based on the 82% figure, despite the acknowledged limitations, such as response bias and social desirability bias.

You should emphasise that while the findings suggest a high prevalence of SVP, the results must be interpreted cautiously, given the study's limitations. A more nuanced statement about the potential overestimation of SVP prevalence would conclude more balanced.

Lack of contextualisation within the Broader Research. It would be best to contextualise the findings within the broader literature by explaining how this study builds on or challenges existing research on SVP. Discuss whether the study introduces new insights or confirms l

I hope this helps to improve the article.

Author Response

Comment: The study addresses an important issue: the Second Victim Phenomenon (SVP) among nurses in Austria. The research appears to be structured appropriately for cross-sectional analysis, clearly focusing on identifying prevalence, symptomatology, and preferred support measures. However, some aspects of the text could be refined to enhance scientific rigour.

Response: Thank you very much for your valuable feedback. We are confident that your suggestions will significantly enhance the scientific quality and clarity of our manuscript. We look forward to incorporating the proposed revisions.

Comment: Introduction - The introduction could more explicitly articulate the knowledge gap. While it mentions studies in Germany and Austria, it does not clearly explain what is still unknown about SVP among Austrian nurses or why this study is essential.

Response: Thank you for your suggestion. To the best of our knowledge, this is the first study conducted across Austria that specifically investigates the prevalence of Second Victims among nurses. We clarified this in the introduction to highlight the importance and uniqueness of our research in addressing this knowledge gap. Examples in the manuskript: “As the first study of this kind”, “Although the second victim phenomenon has been widely studied in other con-texts, no research has yet focused on its prevalence among Austrian nurses. The Se-ViD-A2 study, conducted by the Austrian Second Victim Association and the Wiesbaden Institute for Healthcare Economics and Patient Safety (WiHelP), in collaboration with the Austrian Nurses Association (ÖGKV), seeks to fill this gap by evaluating the prevalence and symptomatology of second victim phenomenon (SVP) among Austrian nurses.”

Comment: Moreover, while it references the history of SVP and some existing studies, the introduction could benefit from more recent and broader literature citations highlighting the current state of SVP research in other countries, mainly focusing on nurses.

Response: We incorporated additional recent literature to provide a more comprehensive overview of the current state of SVP research. We expanded the literature on the Second Victim Phenomenon conducted outside Germany and Austria, further enriching the manuscript’s scholarly content through international literature from the US, Canada, Croatia, Australia, Belgium and Spain (e.g. Goncharuk et al., Wolf et al., Harrison et al., Scott et al., Mira et al.).

Comment: Materials and Methods section - The description of the methodology lacks sufficient detail. There is no clear explanation of how the adaptive questionnaire was designed, nor is there information on its validation specifically for the Austrian context. You provide more details on developing and validating the questionnaire for the Austrian population, even if the instrument was adapted from previous versions. Explain how the questions were culturally relevant and valid for this population. Additionally, it would be helpful to describe the validation process of the adaptive questionnaire.

Response: Thank you for your insightful feedback. The content validation of the questionnaire is thoroughly explained in a German publication, where it was validated for a similar context. In consultation with Austrian experts, we ensured that this validation also applies to the Austrian setting. We adapted the questionnaire to reflect local conditions, language nuances, and demographic context to ensure its cultural relevance and validity for the Austrian population. We included these details in the methodology section to provide greater clarity on the adaptation and validation process. Example in the text: “The questionnaire was adapted to reflect local conditions, language nuances, and demographic context to ensure its cultural relevance and validity for the Austrian population.”

Comment: There is no mention of whether reminders were sent to improve the response rate, nor is it clear how many responses were considered valid from the 6,000 invitations. Moreover, there is no mention of inclusion or exclusion criteria for participating in the study. This raises concerns about whether the sample is representative of the target population. You specify the inclusion and exclusion criteria (if any) for participants. For example, was a minimum number of years of nursing experience required? Are we only those in active practice included?

Response: All registered nurses were invited to participate on a voluntary basis, and the survey was distributed exclusively to nursing staff. The survey was distributed via email to members of the Austrian Nurses Association. We did not employ specific inclusion criteria, as anyone in the profession could potentially become a Second Victim. The only exclusion criterion was a lack of willingness to participate. We included another sentence in 2.3 to clarify this.

Comment: In the statistical analysis section, you should clarify the distinction between ordinal and nominal variables and ensure they are correctly analysed and reported.

Response: We provided a detailed description of the scale levels for the variables and the corresponding reporting methods. Reporting frequencies and percentages offers a simple and clear summary that describes the distribution of responses without making assumptions about the underlying distances between categories. This consistent approach avoids overinterpreting the data while ensuring clear, comparable presentations for categorical data, regardless of whether the variables are nominal or ordinal.For interval-scaled variables, we reported the mean (M) as the measure of central tendency and the standard deviation (SD) as the measure of dispersion, regardless of the distribution characteristics, as discussed in previous research according to Lydersen [1-3]. These metrics were chosen because they provide a clear understanding of the data's central tendency and variability, even when the distribution is skewed.

Comment: The binary logistic regression includes several predictors (e.g., gender, age, personality traits). Still, the relationship between these predictors and the dependent variable (SV experienced at least once: yes vs. no) may not be linear or direct.

Response: We agree, that in binary logistic regression, the assumption is not that the predictors have a direct linear relationship with the outcome but rather that they affect the log-odds of the outcome.Thus, the relationship is not linear in the original outcome but linear in the log-odds.

Comment: You should explain more precisely why bootstrapping was chosen to address any assumptions not met by standard parametric tests (e.g., non-normality of residuals, small sample sizes). Include a rationale for using 5000 bootstrap samples and how this choice affects the stability f the estimates.

Response: By using bootstrapping, researchers can perform various statistical procedures and derive robust parameter estimates, confidence intervals, and standard errors[4]. One of the key benefits is that it allows the application of statistical methods even when assumptions like normal data distribution are not met, as it does not rely on any specific underlying distribution[5, 6]. However, bootstrapping cannot compensate for small sample power. But we had no problem with power in our analyses due to the large sample size.Respecting the views of prominent statistical authorities, it is deemed counterproductive to report post hoc sample power[7]. Consequently, we have chosen to adhere to the recommendation of focusing on confidence intervals (CIs). However, bootstrapping is a broad term that encompasses different resampling techniques. In this study, we used the bias-corrected and accelerated (BCa) method, developed by Efron in 1987[4]. This approach adjusts for bias and skewness in the bootstrapped distributions, leading to more accurate estimates and confidence intervals. As a result, some scholars highly recommend this method. We applied bootstrapping to binary logistic regression, multiple linear regression, and mediation analysis. Choosing to use 5000 bootstrap samples is a common practice that balances computational efficiency with the precision of estimates. By using a large number of bootstrap samples, such as 5000, the estimates derived from the bootstrapping procedure tend to be more stable and reliable. This is because increasing the number of bootstrap samples reduces the variability of the estimates and provides a more accurate representation of the sampling distribution of the statistic of interest. In mediation analysis, bootstrapping is often considered the preferred method for estimating indirect effects and testing the significance of the mediation pathway. This is because traditional methods, such as the Sobel test, rely on assumptions that may not be met in practice, such as normality of the sampling distribution of the indirect effect. Bootstrapping provides a non-parametric approach to mediation analysis that does not rely on these assumptions. By resampling the data and estimating the indirect effect in each resampled dataset, bootstrapping allows for the construction of confidence intervals and hypothesis testing without requiring specific distributional assumptions. That is a reason why e.g., PROCESS Macro includes bootstrapping automatically when calculating diverse mediational and moderation models [8].

Comment: You should ensure that the assumptions of binary logistic regression are met, such as the linearity of the logit for continuous predictors like age. Additionally, more details on the model diagnostics, such as model fit statistics (e.g., AIC, BIC) and pseudo-R-squared values, should be included to evaluate how well the model explains the variation in the data.

Response: Thank you for your insightful comments. We would like to address your points regarding the assumptions of logistic regression, the use of bootstrapping, and model diagnostics.

  1. Pseudo R2 and model fit

While we acknowledge that the interpretation of pseudo R2 values is not always straightforward in the literature, we have included it in our analysis for completeness. The definition of a 'good' pseudo R² value differs across various disciplines. Although these statistics can be somewhat subjective, they are particularly valuable when assessing and comparing different models applied to the same dataset. Since our study is testing a theoretical model rather than adjusting for competing models, pseudo R2 is provided as an additional information but not the indicator of model quality.

  1. ROC Statistics

In addition to pseudo R2, we report the findings of ROC (Receiver Operating Characteristic) statistics to evaluate the model’s predictive power. The ROC curve provides an assessment of the trade-off between sensitivity and specificity, offering a clear picture of the model’s classification accuracy. To achieve this, we calculated the area under the ROC curve (AUC) using saved propensity scores (i.e., the predicted probabilities from the logistic regression model).

  1. Bootstrapping for robust estimation

As mentioned previously, we chose to rely on bootstrapping to provide robust parameter estimates and confidence intervals. Bootstrapping does not require assumptions like linearity in the logit or normality of residuals, which makes it particularly suitable for our data. The resampling process allowed us to obtain reliable estimates even when some assumptions may not have been perfectly met.

  1. No need for AIC/BIC in single-model analysis

We understand your suggestion to report AIC/BIC for model fit evaluation. However, since our analysis is based on a single logistic regression model and not a comparison of multiple models, reporting AIC/BIC is not appropriate in this context. Our primary goal is to assess the theoretical model we tested rather than to compare or optimize between different models. Additionally, if the suggestion was to report AIC/BIC before and after bootstrapping, we would like to clarify that SPSS does not provide AIC/BIC after bootstrapping. More importantly, there is a well-established body of literature that argues AIC and BIC tend to be significantly smaller after bootstrapping, particularly with non-parametric methods[9]. This reduction is often misleading, and some researchers advise against reporting AIC/BIC after bootstrapping because it can be considered unnecessary, as bootstrapping fundamentally alters the basis on which these fit indices are calculated [10]. Given these considerations, we have focused on using bootstrapping for robust parameter estimation without relying on AIC/BIC, which is more suited for model comparison or selection scenarios rather than single-model evaluation. We have included a new subsection within the statistical analysis section that elaborates on these aspects.

Comment: Discussion - The discussion mentions that personality traits had no significant correlation with SV status, except for agreeableness about event type. This is a surprising result but is not discussed in depth. The discussion should also reflect the restricted variance in personality traits mentioned earlier.

Response: In the discussion section, we addressed the lack of significant correlation between personality traits and SV status, with the exception of agreeableness in relation to the type of event.

Comment: The gender disparity finding (that women are more likely to be Second Victims) is mentioned, but the explanation provided (work environment or gender-specific traits) is relatively vague. It would be best to clarify how gender-specific factors may contribute to this disparity—more concrete suggestions.

Response: Thank you for your observation. This is the second study in which we have identified gender as a potential influencing factor. However, most other studies have not found gender to be a significant factor. Therefore, we are cautious in interpreting this finding, as there is doubt about whether it represents a stable influence.

Comment: The failure to identify significant mediation effects is attributed to restricted variance in the data. However, this brief explanation does not fully address potential methodological or theoretical reasons for the lack of mediation effects. You should provide a more in-depth discussion of why the mediation analysis might have failed. Consider discussing whether the theoretical model was appropriately specified or if there were methodological challenges (e.g., insufficient power). Suggest how future studies could be designed to test better mediation effects (e.g., larger sample sizes, different measures).

Response: In the discussion, we explained the origins of the theoretical model and thoroughly examined the lack of significant mediation effects.

Comment: The discussion highlights workplace violence as a primary driver of SVP. Still, it lacks a critical analysis of how systemic issues in the healthcare system may contribute to this phenomenon beyond violence. It would be best if you expanded the discussion to systemic problems in healthcare, such as staffing shortages, high patient loads, and the stressful nature of nursing work. These factors likely contribute to the high prevalence of SVP and should be addressed alongside the discussion on workplace violence.The conclusion draws strong generalisations about the pervasiveness of SVP based on the 82% figure, despite the acknowledged limitations, such as response bias and social desirability bias.

Response: We recognize the importance of addressing systemic issues in healthcare that contribute to SVP, beyond workplace violence. In response to your suggestion, we have expanded the discussion to include an analysis of factors such as staffing shortages, high patient loads, and the stressful nature of nursing work. We believe that these systemic problems significantly impact the prevalence of SVP and warrant thorough examination alongside discussions of workplace violence.Additionally, we have revised the conclusion to clarify that while the 82% figure highlights the significant presence of SVP, we also acknowledge the study's limitations, including response bias and social desirability bias. We have emphasized that these factors may influence the generalizability of our findings.

Comment: You should emphasise that while the findings suggest a high prevalence of SVP, the results must be interpreted cautiously, given the study's limitations. A more nuanced statement about the potential overestimation of SVP prevalence would conclude more balanced.

Response: Thank you for your thoughtful suggestion. While we acknowledge the importance of cautious interpretation, it is worth noting that other studies have reported similarly high prevalence rates of Second Victim Phenomenon (SVP). Additionally, in a later study, we even identified "hidden" Second Victims—individuals who did not self-identify as Second Victims but exhibited corresponding symptoms, which is why we think that the number could be even significantly higher. With the sentence “In light of these limitations, the findings of this study should be interpreted with caution […]” we tried to provide a balanced conclusion, while still addressing the study’s limitations.

Comment: Lack of contextualisation within the Broader Research. It would be best to contextualise the findings within the broader literature by explaining how this study builds on or challenges existing research on SVP. Discuss whether the study introduces new insights or confirms l

Response: Thank you for your feedback. In our discussion section, we recognize the need to contextualize our findings within the broader literature on the SVP. This study builds upon existing research by confirming the prevalence of SVP among healthcare workers while also introducing new insights, particularly regarding the role of the type of critical incident experienced.

Comment: I hope this helps to improve the article.

Response: Thank you for your guidance and support throughout this process. Your feedback has been invaluable in enhancing the depth and clarity of our research.

Literature cited in Revision:

  1. Lydersen, S., Mean and standard deviation or median and quartiles? Tidsskrift for Den norske legeforening, 2020.
  2. Lydersen, S., Statistical review: frequently given comments. Ann Rheum Dis, 2015. 74(2): p. 323-5.
  3. Lydersen, S., How to summarise ordinal data. Tidsskr Nor Laegeforen, 2020. 140(12).
  4. Efron, B., Bootstrap Methods: Another Look at the Jackknife, in Breakthroughs in Statistics: Methodology and Distribution, S. Kotz and N.L. Johnson, Editors. 1992, Springer New York: New York, NY. p. 569-593.
  5. Beasley, W. and J. Rodgers, Bootstrapping and Monte Carlo methods. 2012. p. 407-425.
  6. Chernick, M.R., Bootstrap methods: A guide for practitioners and researchers. 2011: John Wiley & Sons.
  7. Mansournia, M.A., et al., A CHecklist for statistical Assessment of Medical Papers (the CHAMP statement): explanation and elaboration. Br J Sports Med, 2021. 55(18): p. 1009-1017.
  8. Hayes, A.F. PROCESS : A Versatile Computational Tool for Observed Variable Mediation , Moderation , and Conditional Process Modeling 1. 2012.
  9. Brydon, H., R. Blignaut, and J. Jacobs, A weighted bootstrap approach to logistic regression modelling in identifying risk behaviours associated with sexual activity. Sahara j, 2019. 16(1): p. 62-69.
  10. Luus, R., A. Neethling, and T. De Wet, Effectiveness of weighting and bootstrap in the estimation of welfare indices under complex sampling: theory and methods. South African Statistical Journal, 2012. 46(1): p. 85-114.

Reviewer 2 Report

Comments and Suggestions for Authors

Dear Respectable Authors

Thank you for considering this great area of nursing research. You investigated the prevalence, symptomatology, and preferred support measures for SVP among Austrian nurses. Your results are of interest but your manuscript needs some revisions as follows;

- Abstract, please remove subheadings from your abstract.

- Abstract, please add the data collection period.

- Abstract, please add your sampling method and how you determined the sample size.

- Please add the response rate to the abstract and main text.

- Abstract, please add demographic information to the abstract including mean age and gender percentage.

- Keywords, please remove duplicate keywords and replace them with MeSH terms. 

- Please add a comprehensive aim of the study at the end of the introduction section. 

- Methods, please add details regarding the context/setting and the participants including the eligibility criteria.

- Discussion, please add some reasons for why the factor associated with SVP is different between the participants, and how these different factors affect your results.

- Discussion, please add some practical recommendations based on your results for the future research. 

Cheers

Author Response

Comment: Thank you for considering this great area of nursing research. You investigated the prevalence, symptomatology, and preferred support measures for SVP among Austrian nurses. Your results are of interest but your manuscript needs some revisions as follows.

Response: Thank you very much for your thoughtful and constructive feedback. We truly appreciate the time and effort you have taken to provide these valuable suggestions. Your insights have been instrumental in helping us improve the clarity and rigor of our manuscript. We have addressed each of your comments in detail and incorporated the necessary revisions in the updated version of the manuscript, including adjustments to the abstract, methodology, and discussion sections. We believe these changes have strengthened the overall quality of our work, and we look forward to your further feedback.

Comment: Abstract, please remove subheadings from your abstract.

Response: We removed the subheadings

Comment: Abstract, please add the data collection period.

Response: We added the data collection period

Comment: Abstract, please add your sampling method and how you determined the sample size.

Response: All registered nurses were invited to participate on a voluntary basis, and the survey was distributed exclusively to nursing staff. The survey was distributed via email to members of the Austrian Nurses Association. We did not employ specific inclusion criteria, as anyone in the profession could potentially become a Second Victim. The only exclusion criterion was a lack of willingness to participate. We included another sentence in 2.3 to clarify this.

Comment: Please add the response rate to the abstract and main text.

Response: We added the response rate.

Comment: Abstract, please add demographic information to the abstract including mean age and gender percentage.

Response: We added this.

Comment: Keywords, please remove duplicate keywords and replace them with MeSH terms

Response: We deleted two keywords and added MeSH terms

Comment: Please add a comprehensive aim of the study at the end of the introduction section. 

Response: The aim is now written at the end of the introduction (last two paragraphs).

Comment: Methods, please add details regarding the context/setting and the participants including the eligibility criteria.

Response: As mentioned above, all registered nurses were invited to participate on a voluntary basis, and the survey was distributed exclusively to nursing staff. We tried to make it more clear in the methods.

Comment: Discussion, please add some reasons for why the factor associated with SVP is different between the participants, and how these different factors affect your results.

Response: We have addressed these points in our responses to Reviewer 1 and elaborated on them in the discussion section. Here, we describe our findings in detail, exploring the possible causes of the observed differences and offering interpretations within the context of existing literature.

Comment: Discussion, please add some practical recommendations based on your results for the future research.

Response: We added some practical recommendations at the end of the discussion section.

Reviewer 3 Report

Comments and Suggestions for Authors

I congratulate the author, her tutor and advisor for their commitment to the development of research, while reiterating that my observations are only intended to encourage reflection and contribute to improving the study.

In general, I suggest carrying out a grammar and spelling review of the material, as well as as a review of Vancouver standards. Please note that some paragraphs are not indicated of references, and it is necessary to add them. I suggest including more current references and avoiding long paragraphs without fragmentation into periods

Comments on the Quality of English Language

I congratulate the author, her tutor and advisor for their commitment to the development of research, while reiterating that my observations are only intended to encourage reflection and contribute to improving the study.

In general, I suggest carrying out a grammar and spelling review of the material, as well as as a review of Vancouver standards. Please note that some paragraphs are not indicated of references, and it is necessary to add them. I suggest including more current references and avoiding long paragraphs without fragmentation into periods

Author Response

Comment: I congratulate the author, her tutor and advisor for their commitment to the development of research, while reiterating that my observations are only intended to encourage reflection and contribute to improving the study. In general, I suggest carrying out a grammar and spelling review of the material, as well as as a review of Vancouver standards. Please note that some paragraphs are not indicated of references, and it is necessary to add them. I suggest including more current references and avoiding long paragraphs without fragmentation into periods

Response: Dear Reviewer, thank you for your kind words and for your constructive observations. We appreciate your encouragement and your commitment to enhancing the quality of our research. We have conducted a thorough grammar and spelling review of the manuscript, as well as ensured adherence to Vancouver standards. We have also added references where necessary and included more current literature to support our findings. Your feedback has been invaluable in guiding these revisions, and we hope the updated manuscript reflects these improvements.

Round 2

Reviewer 1 Report

Comments and Suggestions for Authors

Dear Authors,

Thank you for submitting your study on such an important topic. I have reviewed the revised manuscript, and the authors have substantially improved the manuscript, effectively addressing most comments. Specifically, comments 1, 2, and 3 have been addressed appropriately in the manuscript revision. Only the first comment might benefit from a more direct reference to how scientific rigour has been improved. Still, the authors have made substantial and detailed improvements to the manuscript.

The authors have provided comprehensive and well-justified responses to most methodological comments. They successfully clarified the validation of the questionnaire, how variables were analysed, and justified the use of bootstrapping.

The main areas that could benefit from more explicit responses are the use of reminders (Comment 5), the depth of discussions on the correlation between personality traits and SVP (Comment 10), gender disparity (Comment 11), and the lack of mediation effects (Comment 12).

Comment 5: The authors state that all registered nurses were invited to participate voluntarily through the Austrian Nurses Association, and they mention that there were no specific inclusion criteria apart from the willingness to participate. The only exclusion criterion mentioned is the lack of desire to participate. However, the authors do not explicitly say whether reminders were sent to improve response rates, which was a part of the reviewer’s comment. Conclusion: The inclusion/exclusion criteria are clarified, but the response could be enhanced by explicitly addressing the use of reminders to increase the response rate.

Comment 10: Authors’ Response: The authors mention that they have addressed the lack of significant correlation between personality traits and SVP, except for agreeableness, in the discussion. However, they do not provide details in the response about how deeply they have explored this in the revised manuscript. Conclusion: This response is relatively vague and could benefit from more specific information about how the discussion has been expanded.

Comment 11: Authors’ Response: The authors explain that they have found gender to be a potential influencing factor in their studies, but this has not been consistently identified as significant in another research. They express caution in interpreting this finding but do not provide a detailed explanation of gender-specific factors contributing to this disparity. Conclusion: While the response acknowledges the finding, a more in-depth exploration of gender-specific factors in the discussion would strengthen the response.

Comment 12: Author’s Response: The authors explain that the theoretical model was introduced in the discussion and have thoroughly examined the reasons for the lack of significant mediation effects. However, as with Comment 10, the response does not detail the exact changes made in the manuscript.

Conclusion: This response is acceptable, but it could benefit from more details about the specific changes made to the discussion.

Best regards

Author Response

Comment: The main areas that could benefit from more explicit responses are the use of reminders (Comment 5), the depth of discussions on the correlation between personality traits and SVP (Comment 10), gender disparity (Comment 11), and the lack of mediation effects (Comment 12).

Response: Thank you very much for your thorough review and thoughtful comments. We greatly appreciate your positive feedback on the revisions made and the opportunity to further improve the manuscript. In response to the remaining areas of concern, we will address the use of reminders (Comment 5), the discussion on personality traits and SVP (Comment 10), gender disparity (Comment 11), and the lack of mediation effects (Comment 12) in more detail below.

Comment 5: The authors state that all registered nurses were invited to participate voluntarily through the Austrian Nurses Association, and they mention that there were no specific inclusion criteria apart from the willingness to participate. The only exclusion criterion mentioned is the lack of desire to participate. However, the authors do not explicitly say whether reminders were sent to improve response rates, which was a part of the reviewer’s comment. Conclusion: The inclusion/exclusion criteria are clarified, but the response could be enhanced by explicitly addressing the use of reminders to increase the response rate.

Response: Thank you for your comment. We did not send any reminders, as the response rate progressed rapidly and was satisfactory without additional prompts.

Comment 10: Authors’ Response: The authors mention that they have addressed the lack of significant correlation between personality traits and SVP, except for agreeableness, in the discussion. However, they do not provide details in the response about how deeply they have explored this in the revised manuscript. Conclusion: This response is relatively vague and could benefit from more specific information about how the discussion has been expanded.

Response: Thank you very much for your valuable comment. We are pleased to point out the relevant passage that we added in the last revision, which addresses this point in more detail:

  • Line 449-461: “Nevertheless, the emergence of agreeableness as significant only after accounting for the types of adverse events highlights its context-dependent role in SV development. This suggests that agreeableness does not universally predispose individuals to SV; rather, it becomes relevant in specific situations involving interpersonal dynamics or emotional sensitivity [45]. Typically associated with empathy and cooperation, agreea-bleness may lead to greater emotional distress in events involving patient harm, where an individual’s desire for harmony is challenged [45]. The finding that agreeableness was not significant any more after event types were considered implies that it may in-teract with the nature of adverse events to indirectly influence SV outcomes. This em-phasizes the necessity of considering both personality traits and specific event circum-stances when predicting SV. To summarize, the relationship between agreeableness and event type was noteworthy, but this finding should be interpreted cautiously given the restricted range of personality scores, which could limit the generalizability of the re-sults.”
  • Line 497-505: “While certain traits, like neuroticism and agreeableness, are frequently discussed in the literature as being influential in the context of stress and emotional well-being, our findings suggest that their predictive power may depend on specific circumstances or additional moderating factors. This appears to be the case for agreeableness, which lost its significance once the type of adverse event was included in the equation as previ-ously noted. In light of these findings, we recommend that future research includes a longitudinal design and broader demographic, particularly older healthcare workers, and collection of further contextual variables to explore how age and experience corre-late with personality traits and symptom load.”

Comment 11: Authors’ Response: The authors explain that they have found gender to be a potential influencing factor in their studies, but this has not been consistently identified as significant in another research. They express caution in interpreting this finding but do not provide a detailed explanation of gender-specific factors contributing to this disparity. Conclusion: While the response acknowledges the finding, a more in-depth exploration of gender-specific factors in the discussion would strengthen the response.

Response: We are able to reference passages in the manuscript that were included in the last revision, and we have also added one sentence to clarify.

  • Line 466-471: “The persistent finding that women are at a higher risk of becoming SVs aligns with some previous research [16] but stays in contrast to a previous survey among German nurses [19]. It raises questions about underlying causes. The gender disparity in SVP risk may be influenced by specific work environments and roles predominantly occupied by men and women, suggesting that job-specific factors and gender-specific personality traits both play significant roles in susceptibility to SVP [47,48].”
  • Additionally, we tried to clarify gender-specific factors with this sentence in line 471-474: “Gender-specific factors, such as differences in emotional labor and coping mechanisms may contribute to the observed disparity, but further research is needed to fully understand the extent and nature of these influences.”

Comment 12: Author’s Response: The authors explain that the theoretical model was introduced in the discussion and have thoroughly examined the reasons for the lack of significant mediation effects. However, as with Comment 10, the response does not detail the exact changes made in the manuscript.

Response: Thank you very much for your comment. We would like to direct you to the relevant passage in the manuscript, which was included in the previous revision to address this point:

  • line 475-485: “(The study's failure to identify significant mediation effects could be attributed to restricted data variance due to younger sample.) In our analysis of the Sevid-III study, we found that years of job experience were no longer significant predictors of symptom load following a second victim (SV) event after accounting for neuroticism in the mul-tiple linear regression. This led us to conduct an inductive, bottom-up mediation anal-ysis, which revealed a negative indirect effect of years of experience on symptom load through neuroticism. Similar findings were noted in the Sevid-IX and Sevid-A1 studies. We interpreted this negative indirect effect as potentially indicative of a self-selection. Specifically, individuals with higher neuroticism may have opted to leave their posi-tions or change careers, while those with lower neuroticism remained, resulting in lower neuroticism scores among the more experienced workers.”
  • Line 485-495: “This could also mean that more experienced employees benefit from stronger social networks and access to emotional support at work, further mitigating their symptom load. Additionally, recent research into the baby boomer generation suggests that neuroticism tends to decline with age [49]. Since age correlates with years of experience, we posited that this decline could lead to lower neuroticism and, consequently, reduced symptom load. However, our study found no direct correlation between neuroticism and symptom load, which hindered the manifestation of a significant indirect effect. Although the theoretical framework is sound and neuroticism serves as a mediating variable in various contexts, including the Job Demands-Resources model [50,51], the specific characteristics of our sample may not have provided enough variability in neuroticism to manifest a clear mediation effect.”

We hope that our responses adequately address your concerns and further enhance the quality of our manuscript. Thank you once again for your insightful feedback.